

# Validation of the "Quality of Life in School" instrument in Canadian elementary school students

Satvinder Ghotra[1], Jessie-Lee D. McIsaac[2], Sara F.L. Kirk[2] and Stefan Kuhle[3]

[1] Department of Pediatrics, Division of Neonatal-Perinatal Medicine, Dalhousie University, Halifax, NS, Canada
[2] School of Health and Human Performance, Dalhousie University, Halifax, NS, Canada
[3] Department of of Pediatrics and Obstetrics & Gynaecology, Dalhousie University, Halifax, NS, Canada

## ABSTRACT

**Background.** School is an integral component of the life of a child, and thus quality of school life is an important part of the overall quality of life experienced by a child. There are a few instruments available to measure the quality of school life but they are often not available in English, or they are not appropriate for use alongside other instruments in a survey of young children. The Quality of Life in School (QoLS) instrument is a short, self-report measure to assess elementary school students' perception of their quality of school life in four domains. The instrument was developed in Israel and has been validated among Hebrew-speaking children. The aim of the current study was to evaluate the psychometric properties of the QoLS measure in Canadian elementary school children.

**Methods.** A total of 629 children attending grades 4–6 were recruited in a population-based cross-sectional study. The QoLS measure was administered to participating children by trained research assistants. In addition, their socio-demographic details and academic data were also obtained. The psychometric testing included exploratory factor analysis and reliability estimation using internal consistency (Cronbach's Alpha). Construct validity was investigated using the known groups comparisons for discriminative validity and via convergent validity.

**Results.** A four-factor structure was generated explaining 39% of the total variance in the model. The results showed good internal consistency and acceptable floor and ceiling effects. Cronbach's Alpha ranged from 0.75 to 0.93. Known groups comparisons showed that the QoLS measure discriminated well between subgroups on the basis of gender, grade, and academic achievement, thus providing evidence of construct validity. The convergent validity was also appropriate with all the four domains demonstrating moderate to strong correlations to each other and to the total QoLS score.

**Conclusions.** QoLS appears to be a valid and reliable measure for quality of school life assessment in young Canadian children.

Corresponding author
Stefan Kuhle, stefan.kuhle@dal.ca

## INTRODUCTION

Quality of life has been acknowledged as an important and relevant concept for people of all ages including school-aged children. In children, the concept of quality of life arises

from a dynamic interaction of several variables including child, family, environment and school. Of these, school is an integral component of the life of a child, and thus quality of school life is an important part of the overall quality of life experienced by a child. Quality of school life could be defined as well-being resulting from children's integration into the life and the environment of their schools (*Karatzias et al., 2001*) and represents the degree of satisfaction or dissatisfaction perceived by children with their school life (*Epstein & McPartland, 1976*). Quality of school life has been conceived as an important aspect of schooling by numerous authors since it is associated with students' academic motivation and performance (*Karatzias, Power & Swanson, 2001*; *Mok & Flynn, 2002*; *Goodenow & Grady, 1993*; *Murray-Harvey, 2010*; *Wang & Holcombe, 2010*). Quality of school life can be viewed as a measure of students' attitude and feelings towards school, which in turn is positively associated with their intentions to continue at school (*Ainley, Foreman & Sheret, 1991*).

Various factors may potentially influence quality of school life, such as the age and gender of students. The literature on gender influences on quality of school life has shown conflicting results. Many studies reported that girls perceive a better quality of school life compared to boys (*Mok & Flynn, 2002*; *Malin & Linnakyla, 2001*; *Konu & Lintonen, 2006a*; *Løhre, Lydersen, Vatten, 2010*; *Kong, 2008*), while others found no differences between the genders (*Gillman & Huebner, 2006*; *Weintraub & Bar-Haim Erez, 2009*). However, the effect sizes in some of the studies that found significant gender differences were often very small (<0.2 standard deviations) (*Mok & Flynn, 2002*; *Konu & Lintonen, 2006a*) or the gender differences were not consistently found for all domains of an instrument (*Konu & Lintonen, 2006a*). The reasons for the higher perceived quality of school life in girls may be a better match of the female gender role with school expectations (*Samdal et al., 1998*). Girls also appear to have a more positive view of teacher support and the school environment (*Konu & Lintonen, 2006a*). By contrast to gender, there is consensus in the literature that age is an important determinant of quality of school life with younger students perceiving their quality of school life higher than their older peers (*Kong, 2008*; *Weintraub & Bar-Haim Erez, 2009*; *Samdal et al., 1998*; *Konu & Lintonen, 2006a*; *Park, 2005*). With increasing age in childhood and adolescence, life satisfaction decreases across all areas (i.e., family, friends, school, living environment) (*Park, 2005*).

In Canada, 98% of children are in the school system for almost half their waking hours and schooling is compulsory from age five to sixteen (with minor variations dependent on the province/territory). The Health Behaviour in School-aged Children (HBSC) is a cross-national survey of school students that collects data every four years on students' health and well-being, social environments and health behaviours (*HBSC, International Coordinating Centre*). The survey includes several items related to quality of school life, including feelings of school satisfaction, belonging and safety and perceptions of teachers and peers. Although a only minority of students reported liking school a lot, the majority of students reported that they felt they belonged at their school. The HBSC results in Canada are similar to the international results previously reported with differences reported across gender and age (e.g., students' positive perceptions of teachers decreased with grade level with girls being more likely to report positive perceptions) (*Freeman, King & Pickett, 2011*).

There are a number of tools available for quality of school life assessment in young children but many of them are not available in English, do not cover all domains of quality of school life, or are impractical for use in a research study (*Murray-Harvey, 2010*; *Konu & Lintonen, 2006a*; *Løhre, Lydersen, Vatten, 2010*; *Weintraub & Bar-Haim Erez, 2009*; *Opdenakker & Van Damme, 2000*; *Sari, 2012*; *Rauer & Schuck, 2004*). The majority of research on quality of school life and student well-being instruments has been published in the education research literature, which may make the instruments less accessible to health researchers. The *School Well-Being Profile* was developed and validated for Finnish school children in grades 4–12 as a tool to help schools evaluate well-being of their students (*Konu & Lintonen, 2006a*; *Konu & Lintonen, 2006b*; *Konu et al., 2002*); an English translation is available. The instrument consists of 56 (elementary school) and 80 (high school) questions, respectively, in four domains (*school conditions*, *social relationships*, *self-fulfillment*, *health*). Administration of the *School Well-Being Profile* is web-based, and the tool has been tested and used in a number of settings but to the best of our knowledge is used mostly for school health evaluations, and less for academic research. Its computer-based mode of administration may make it impractical to use in an otherwise paper-based research survey. Also, the large number of items in the *School Well-Being Profile* may place too large a burden on younger students, especially if the measure is used alongside other measures in a comprehensive survey. The *Quality of School Life* (QSL) scale based on the work of *Williams & Batten (1981)* and others (*Ainley & Bourke, 1992*; *Mok, 1992*) was designed to measure the well-being of Australian high school students but there is no simplified version for elementary school students. *Murray-Harvey, (2010)* developed an 86-item *YourLife at School* questionnaire with five domains (supportive relationships, stressful relationships, psychological health, social/emotional adjustment, academic performance) and validated it in Australian school children in grades 5–9. In the Netherlands, *Opdenakker & Van Damme (2000)* constructed a well-being scale consisting of eight indicators (*Well-being at school*, *Social integration in the class*, *Relationship with teachers*, *Interest in learning tasks*, *Motivation towards learning tasks*, *Attitude to homework*, *Attentiveness in the classroom*, and *Academic self-concept*) based on an earlier school well-being measure (*Stoel & Pijl, 1980*). Unfortunately, this instrument and most of the publications on them are only available in Dutch. *Løhre, Lydersen, Vatten, 2010* developed a set of 49 Likert-type and open-ended questions that examine academic success, peer and teacher relationships, and support in school but not the school environment (*Løhre, Lydersen, Vatten, 2010*). The questionnaire was tested in 423 students in grades 1–10 in Norway and showed acceptable test–retest reliability. However, instrument structure, internal consistency, and validity were not formally tested. The *Quality of Life in School* (QoLS) instrument is a self-report measure developed in Israel to assess elementary school students' perception of their quality of school life (*Weintraub & Bar-Haim Erez, 2009*). The QoLS scale is a paper-based instrument that assesses all relevant domains of students' well-being in school including the school environment, while still being brief enough (37 items) to be used with other instruments in a comprehensive survey of young children. For these reasons, we felt that the QoLS instrument was most suitable measure to assess quality of school life and the impact

of interventions improving student health and well-being in our study of elementary school children in Canada.

The QoLS instrument is constructed based on the theoretical definition of quality of school life described by *Malin & Linnakyla (2001)*, and the biopsychosocial model (*Engel, 1977*). As per Malin and Linnakyla's definition, quality of school life is "students' general well-being and satisfaction, from the point of view of their positive and negative experiences, particularly in activities typical of school." The biopsychosocial model places an individual's relationships and activities in the context of their environment. Therefore, an instrument that is based on these two frameworks should not only examine the students' wellbeing at school but also their relationships with their peers and teachers, as well as the school's physical environment and activities. Items for the instrument were developed from the theoretical literature on quality of school life and from semi-structured interviews with students, parents, and teachers. The QoLS instrument has multidimensional structure consisting of four factors: teacher–student relationship and school activities, physical environment, negative feelings toward school, and positive feelings toward school (*Weintraub & Bar-Haim Erez, 2009*). The framework of the QoLS differs from that of the *School Well-Being Profile*, a well-researched and frequently used measure of QSL developed in Finland (*Konu & Lintonen, 2006a*; *Konu & Lintonen, 2006b*; *Konu et al., 2002*). The *School Well-Being Profile* is grounded in Allardt's model of well-being (*Allardt, 1976*), which views well-being as a state that allows an individual to satisfy their basic needs. The model uses three indicator categories for an individual's needs: having, loving, and being. The developers of the *School Well-Being Profile* applied this model to students in schools and defined well-being in school as a four-dimensional construct with the domains school conditions (having), social relationships (loving), means for self-fulfillment (being), and health status (health) (*Konu & Lintonen, 2006a*). While the underlying framework may differ between the *School Well-Being Profile* and the QoLS scale, there are a lot of communalities, and both instruments assess the concepts environment, relationships, attitude, and health, albeit partly in different domains. The QSL scale, originally developed by *Williams & Batten (1981)*, is based on society level imperatives of schooling (adaptation, goal attainment, integration, and latency) that were translated to the corresponding student experiences of schooling (opportunity to learn, relevance or utility of schooling, identification with the student role, and perception of own status as student). The instrument has been refined over the years by a number of researchers (*Ainley & Bourke, 1992*; *Mok, 1992*), and the latter four categories along with a measure of the perception of teachers and two affective outcomes now form the seven subscales of the instrument (General affect, Negative affect, Status, Identity, Teachers, Opportunity, Achievement) (*Mok & Flynn, 2002*). There is some overlap in the items and domains with both the QoLS scale and the School Well-Being Profile but the most notable difference is the absence of the assessment of the school environment in the QSL instrument.

Since the QoLS instrument has not yet been validated in Canadian students, the objective of this study was to determine the reliability and validity of the measure in a sample of elementary school students attending grades 4–6 in the Canadian province of Nova Scotia.

## MATERIALS & METHODS

A cross-sectional population-based study was conducted in spring 2014 to evaluate school health practices and student health and health behaviours in the Tri-County Regional School Board (TCRSB), Nova Scotia, Canada. The TCRSB is a rural school board located at the South-Western tip of Nova Scotia that encompasses the three counties of Shelburne, Yarmouth, and Digby. The Board serves approximately 6,400 students in 27 schools in an area of over 7,000 square kilometers. All schools and students attending grade 4–6 (age 9–12 years) in the school board were invited to participate. The trained research assistants visited these schools and provided all eligible students with the study packages encompassing parental consent forms and a parental survey. Parents were requested to complete these at home and deliver them back to school. Research assistants visited the schools again to gather the parental information and students were invited to complete the QoLS questionnaire. To ensure a uniform assessment and to prevent any bias, the research assistants were trained to administer the survey. Ethics approval for this study was obtained from the Health Research Ethics Boards at the Dalhousie University (Protocol # 2013-3094). The permission for data collection was also granted from the participating school board and individual schools. Written informed consent was obtained from the parents of all children included in the study.

All 18 eligible schools the school board agreed to participate. A total of 1,445 students and their parents were invited to participate, of whom 670 responded resulting in a response rate of 46%. The student survey was filled out by 636 students; of these 18% had missing responses for less than four of the 37 questions, and only seven students (1%) had four or more missing responses and were removed from the analysis, resulting in a final sample size of 629. Parental consent to link the study information with academic performance data was obtained for 565 students. Non-response appeared to be largely driven by competing priorities in schools that resulted in non-compliance of teachers in reminding students to return their consent forms.

### QoLS questionnaire

The QoLS questionnaire is a self-report measure of students' quality of school life developed in the Hebrew language (*Weintraub & Bar-Haim Erez, 2009*). The measure consists of 36 items organized into 4 domains: *teacher–student relationship and social activities* (12 items), *physical environment* (11 items), *negative feelings towards school* (8 items) and *positive feelings towards school* (5 items). The participants respond on a 4-point Likert scale (from "never true" to "always true"). Each item is scored on a scale from 1 to 4 with negative items being reverse scored. The instrument provides four domain scores and a total QoLS score, with higher scores indicating better quality of school life. The scale has been demonstrated to be reliable and valid in Hebrew-speaking elementary school age children in Israel (*Weintraub & Bar-Haim Erez, 2009*). The current study used the English translation (by its creators) of the QoLS with some minor changes to the wording of some of the items (by the authors), e.g., "whiteboard" instead of "blackboard," or "walk/bike/ride to school" instead of "way to school"; the English version used contained an additional

item (classroom lighting) that was later dropped from the Hebrew version of the scale. Completion of the scale questions by the students took about ten minutes.

## Socio-demographic information

The parent survey contained questions on socio-demographic factors such as household income (4 levels: $0–$40,000; $40,001–$60,000; $60,001–$100,000; >$100,000 CDN), parental education (secondary school or less, college, or university), and area of residence (urban vs. rural; based on postal code). In addition, parents also reported their child's health as perceived by themselves (excellent, very good, good, or poor).

## Academic performance data

The academic performance for Mathematics and English Language Arts (ELA) was also collected for the 2013/14 school year for all students. Grades were obtained directly from the school board for each of the three terms. Grades (ranging from "A" to "D") were transformed to a numeric scale (1–4), and the median of the three terms was used as the overall grade estimate for the academic year.

## Statistical analysis

Descriptive statistics were presented as relative frequencies or means and standard deviations as applicable. Missing QoLS information for students with ≤3 missing responses was imputed using hot deck imputation. Hot deck imputation imputes missing values using non-missing information from randomly selected, similar (here: students in the same school) observations in the same dataset. Scoring of the measure was performed using the standard scoring instructions with negative items being reverse scored. A mean score was computed for all the domains and the overall scale.

Because the QoLS instrument was applied in a culturally different context and in another language, we decided to perform an exploratory factor analysis (with varimax rotation) rather than a confirmatory factor analysis to examine the multidimensional structure of the instrument. The factor analysis was based on a polychoric correlation matrix of the item responses. Eigenvalues, scree plot, and parallel analysis were used to determine the number of factors to retain (*Hayton, Allen & Scarpello, 2004*). Factor loadings of ≥0.30 were used to allocate items to a scale. Items that loaded on more than one factor were assigned to the factor with the higher loading. Content validity of the QoLS was presumed to be appropriate. Internal consistency reliability for the entire questionnaire and each domain was assessed using Cronbach's Alpha, with values between 0.70 and 0.95 considered acceptable.

Construct validity was examined using several *a priori* hypotheses for discriminative and convergent properties of the instrument. Children's QoLS scores were expected to differ by (1) gender (girls will report higher scores than boys), (2) school grade (grade 4 students will perceive higher scores than grade 5 or 6 students), (3) parent report of child's health status (children with better health status will have a higher scores) and (4) academic ranking (children with better academic grades will have higher scores). Comparison between these known groups was performed using *t*-test or ANOVA as applicable. Convergent validity was assessed using Pearson's correlations among different subscales and the overall scale.

**Table 1  Socio-demographic characteristics of the study sample ($n = 629$).**

| | Frequency ($n$) |
|---|---|
| **Male gender** | 48% (301) |
| **Grade** | |
| 4 | 33% (209) |
| 5 | 37% (232) |
| 6 | 30% (188) |
| **Age** | |
| 9 | 12% (77) |
| 10 | 33% (205) |
| 11 | 31% (194) |
| 12 and older | 24% (153) |
| **Residence** | |
| Rural | 65% (411) |
| Urban | 35% (328) |
| **Household income** | |
| Up to $40,000 | 27% (167) |
| $40,001–$60,000 | 14% (90) |
| $60,001–$100,000 | 24% (148) |
| >$100,000 | 13% (82) |
| Missing | 23% (142) |
| **Parental education** | |
| Secondary school or less | 25% (157) |
| College | 48% (302) |
| University | 23% (144) |
| Missing | 4% (26) |

We hypothesized moderate ($r = 0.40$–$0.59$) to strong ($r > 0.60$) correlations between different subscales of QoLS and to the total QoLS, as they measure the same construct. Floor and ceiling effects were considered to be present when more than 15% of the students obtained the lowest or the highest possible scores. The significance level was set at $p < 0.05$. The analysis was performed using R 3.1 (*R Core Team, 2015*) with the *psych* package (*Revelle, 2015*).

## RESULTS

The socio-demographic characteristics of the 629 children are presented in Table 1. The distribution of students among the three grade levels was fairly even. About two-thirds of students resided in urban areas, 31% came from households with an annual household income ≤$60,000 (median total household income in Canada in 2013: $76,550), and 26% of parents and caregivers had secondary school education or less.

The responses to the individual scale questions are shown in Fig. 1. Eigenvalues, scree plot, and parallel analysis suggested the presence of four factors: *Psychosocial* (PS) (eigenvalue: 4.34); *Attitude towards school* (AT) (eigenvalue: 3.67); *School environment* (SE) (eigenvalue: 3.12); and *Teacher–student relationship* (TS) (eigenvalue: 3.12). Item loadings are shown in Table 2 and Fig. 2. There were 34 items in the final scale with 12, 8, 8 and 6 items in PS, AT, SE and TS factors respectively. Three items (trip to school, lighting in classroom, social activities at school) did not load on any of the factors and were

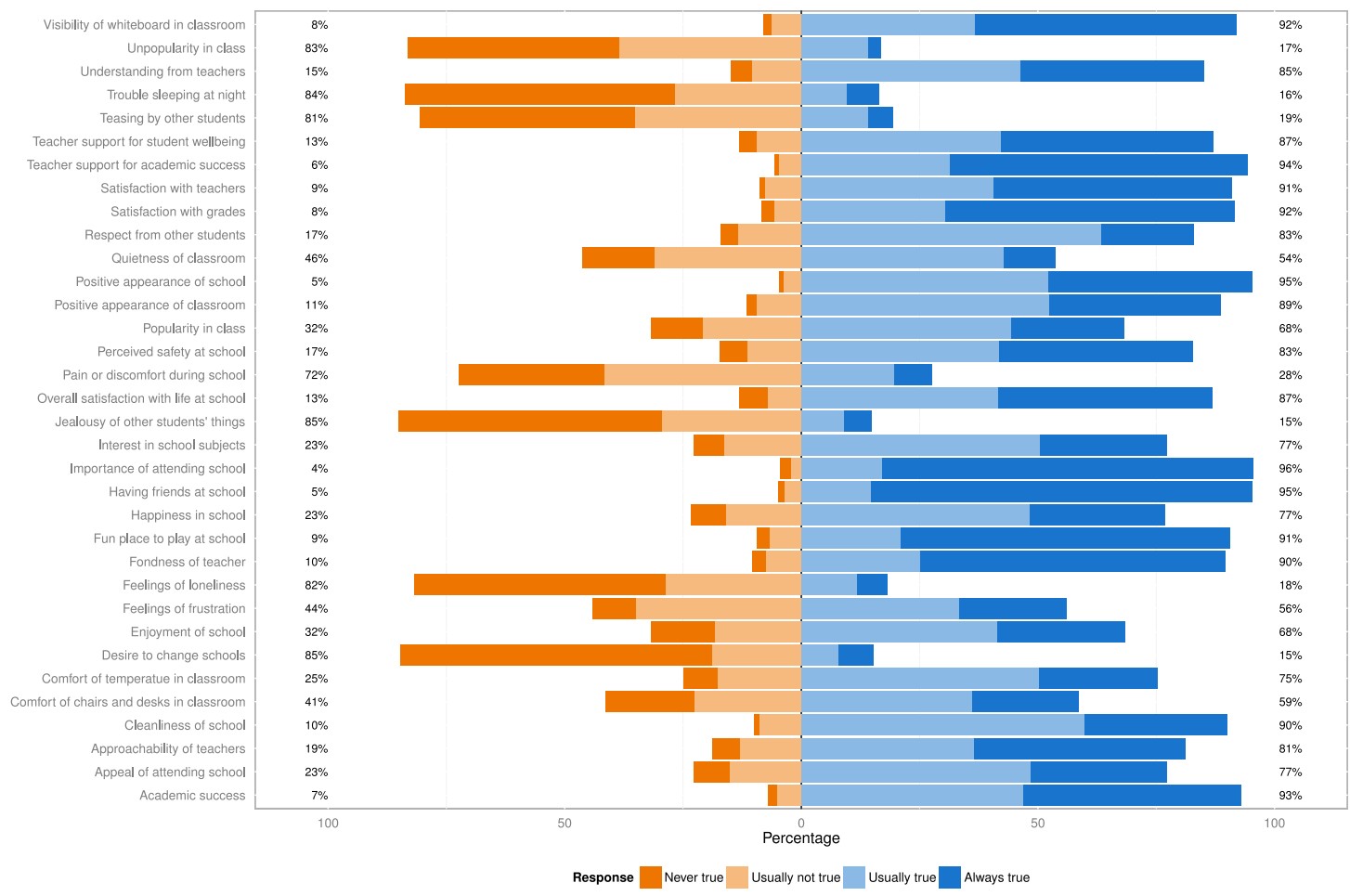

**Figure 1** Responses to the 37 items on the Quality of Life in School (QoLS) scale (*n* = 629).

dropped. The 4-factor structure explained 39% of the total variance in the model. The mean scale scores and other scale characteristics are shown in Table 3. Cronbach's Alpha ranged from 0.75 to 0.93 for the domains. Floor and ceiling effects were 0% for the total QoLS score.

Discriminative properties of the instrument are listed in Table 4. The mean QoLS scores differed by gender with girls reporting having higher scores than boys. The difference was statistically significant in all except the overall score ($P = 0.061$) and the PS domain. Quality of school life also differed by grade with grade 4 students demonstrating significantly higher scores than grade 5 and 6 students in multiple domains. Students achieving higher academic grades had significantly higher scores in all domains of QoLS measure. By contrast, students' quality of school life did not differ significantly by parent report of child's health status except in the PA domain. Correlations between different domains of the instrument were moderate to strong (all $P < 0.0001$) (Table 5). All four domains strongly correlated with the total QoLS score.

**Table 2 Factor loadings for the 37 items on the Quality of Life in School (QoLS) scale.** Last column on the right indicates lower cross-loadings on other factors. Three items with loadings <0.30 were dropped from the analysis (Social activities at school, Lighting in classroom, Trip to school comfortable).

|  | F1 | F2 | F3 | F4 |  |
|---|---|---|---|---|---|
| **F1—Psychosocial** | | | | | |
| Feelings of loneliness | −0.77 | | | | |
| Teasing by other students | −0.62 | | | | |
| Having friends at school | 0.59 | | | | |
| Trouble sleeping at night | −0.59 | | | | |
| Unpopularity in class | −0.57 | | | | |
| Respect from other students | 0.52 | | | | F4* |
| Feelings of frustration | −0.48 | | | | |
| Jealousy of other students' things | −0.47 | | | | |
| Perceived safety at school | 0.45 | | | | |
| Desire to change schools | −0.45 | | | | |
| Popularity in class | 0.44 | | | | |
| Pain or discomfort during school | −0.38 | | | | |
| **F2—Attitude towards school** | | | | | |
| Interest in school subjects | | 0.73 | | | |
| Enjoyment of school | | 0.65 | | | |
| Overall satisfaction with life at school | | 0.62 | | | F1* |
| Happiness in school | | 0.61 | | | F1* |
| Interest in school subjects | | 0.57 | | | F3* |
| Importance of attending school | | 0.46 | | | |
| Satisfaction with grades | | 0.35 | | | |
| Academic success | | 0.34 | | | |
| **F3—Teacher–student relationship** | | | | | |
| Teacher support for student well-being | | | 0.70 | | F4* |
| Fondness of teacher | | | 0.69 | | |
| Understanding from teachers | | | 0.63 | | |
| Approachability of teachers | | | 0.59 | | |
| Teacher support for academic success | | | 0.53 | | F4* |
| Satisfaction with teachers | | | 0.47 | | F4* |
| **F4—School environment** | | | | | |
| Quietness of classroom | | | | 0.55 | |
| Positive appearance of school | | | | 0.55 | |
| Positive appearance of classroom | | | | 0.50 | |
| Cleanliness of school | | | | 0.49 | |
| Comfort of chairs and desks in classroom | | | | 0.45 | |
| Fun place to play at school | | | | 0.43 | |
| Comfort of temperature in classroom | | | | 0.40 | |
| Visibility of whiteboard in classroom | | | | 0.30 | |

## DISCUSSION

Our study assessed the psychometric properties of the QoLS measure in Canadian elementary school children and provided some evidence in favour of its reliability and construct validity. The study findings indicate that the QoLS tool is an acceptable and psychometrically robust measure to assess quality of school life in young Canadian school aged children attending grade 4–6 (aged around 9–12 years). The study findings also suggest that the QoLS measure is potentially suitable for inclusion in large-scale surveys to assess quality of school life in elementary school students.

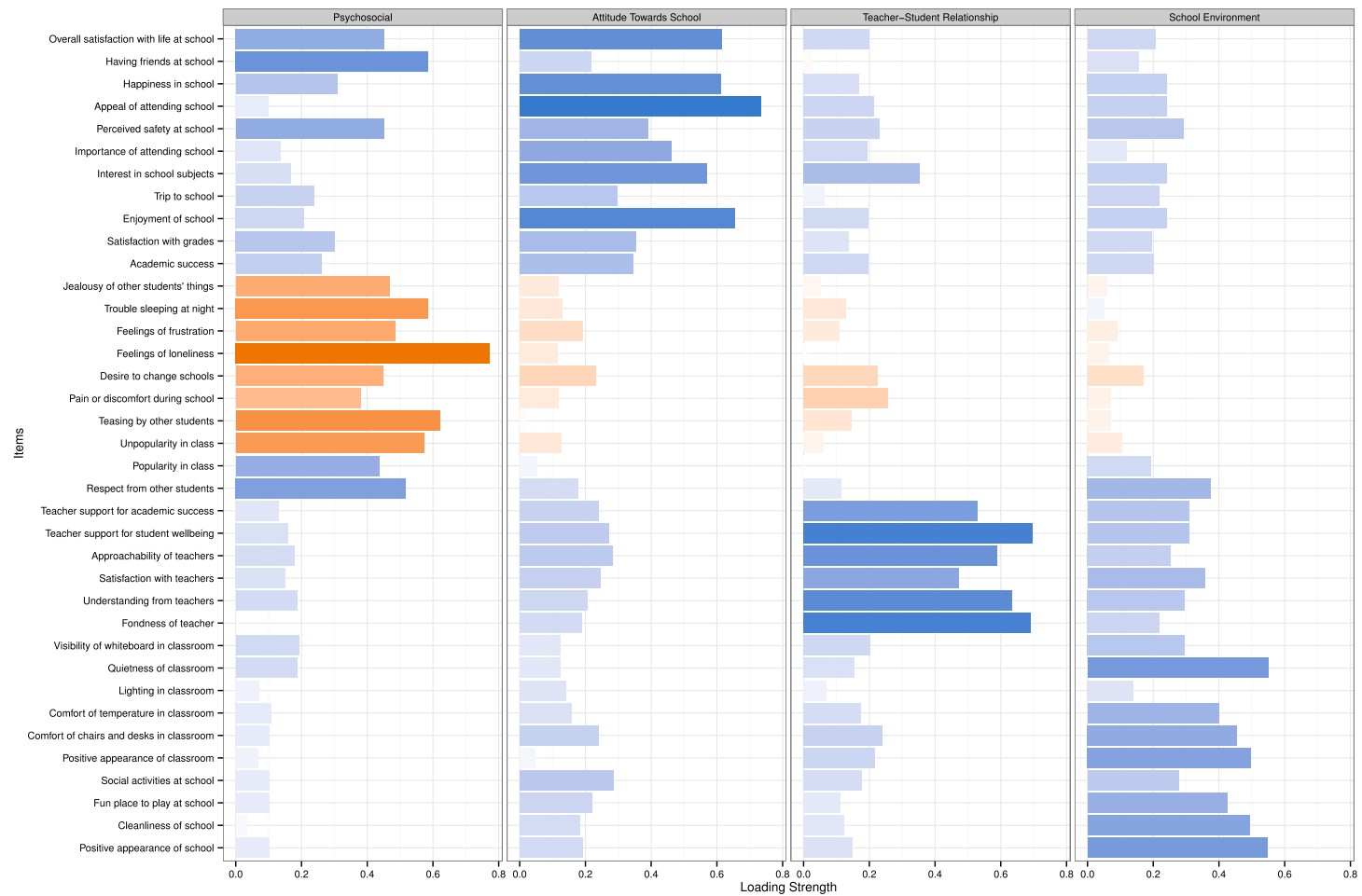

**Figure 2  Factor loadings for the 37 items on the Quality of Life in School (QoLS) scale.** Blue and orange bar colors indicate positive and negative loadings, respectively. Three items with loadings <0.30 were dropped from the analysis (Social activities at school, Lighting in classroom, Trip to school comfortable).

**Table 3  Characteristics of the Quality of Life in School (QoLS) scale in the study sample (*n* = 629).**

| Domains | Score | | | Cronbach's alpha | % floor | % ceiling |
|---|---|---|---|---|---|---|
| | Mean (SD) | Minimum | Maximum | | | |
| Total QoLS | 3.19 (0.43) | 1.50 | 3.94 | 0.93 | 0 | 0 |
| Psychosocial | 3.15 (0.52) | 1.17 | 4 | 0.85 | 0 | 0.3 |
| Attitude towards school | 3.20 (0.58) | 1.12 | 4 | 0.87 | 0 | 6.2 |
| School environment | 3.11 (0.47) | 1.38 | 4 | 0.75 | 0 | 1.4 |
| Teacher–students relationship | 3.36 (0.59) | 1 | 4 | 0.87 | 0.2 | 17.3 |

**Notes.**
Abbreviations: *SD*, Standard deviation.

The factor analysis yielded four subscales (*psychosocial*, *attitude towards school*, *school environment* and *teacher–student relationship*) that are similar but not identical to the four subscales reported for the original QoLS measure. Weintraub and Erez' *Negative feelings toward school* category appears to correspond to the *Psychosocial* domain from our

**Table 4  Discriminative validity: mean scores, standard deviation, effect sizes, and *P* values across different domains of the Quality of Life in School (QoLS) scale by gender, grade, parent report of child health status, and academic performance in English and Mathematics.**

| Variable | n | Total QoLS | | Psychosocial | | Attitude towards school | | School environment | | Teacher–student relationship | |
|---|---|---|---|---|---|---|---|---|---|---|---|
| | | Mean (SD) | ES | Mean (SD) | ES | Mean (SD) | ES | Mean (SD) | ES | Mean (SD) | ES |
| **Gender** | | | | | | | | | | | |
| Female | 328 | 3.22 (0.42) | Ref | 3.14 (0.53) | Ref | 3.27 (0.54) | Ref | 3.15 (0.45) | Ref | 3.41 (0.57) | Ref |
| Male | 301 | 3.16 (0.45) | −0.15 | 3.17 (0.52) | 0.05 | 3.13 (0.61) | −0.25 | 3.07 (0.49) | −0.18 | 3.30 (0.61) | −0.19 |
| *P* value | | 0.061 | | 0.577 | | 0.002 | | 0.027 | | 0.018 | |
| **Grade** | | | | | | | | | | | |
| 4 | 209 | 3.24 (0.42) | Ref | 3.15 (0.53) | Ref | 3.24 (0.59) | Ref | 3.20 (0.44) | Ref | 3.48 (0.52) | Ref |
| 5 | 232 | 3.19 (0.46) | −0.10 | 3.17 (0.52) | 0.04 | 3.19 (0.60) | −0.08 | 3.10 (0.48) | −0.21 | 3.38 (0.61) | −0.19 |
| 6 | 188 | 3.13 (0.42) | −0.26 | 3.14 (0.52) | −0.02 | 3.17 (0.53) | −0.12 | 3.02 (0.47) | −0.38 | 3.20 (0.60) | −0.51 |
| *P* value | | 0.042 | | 0.815 | | 0.502 | | 0.001 | | <0.001 | |
| **Health status** | | | | | | | | | | | |
| Poor | 0 | – | – | – | – | – | – | – | – | – | – |
| Fair | 32 | 3.04 (0.60) | Ref | 3.02 (0.64) | Ref | 2.96 (0.83) | Ref | 2.99 (0.58) | Ref | 3.24 (0.75) | Ref |
| Good | 114 | 3.15 (0.44) | 0.24 | 3.08 (0.53) | 0.12 | 3.15 (0.64) | 0.28 | 3.09 (0.46) | 0.20 | 3.37 (0.59) | 0.21 |
| Very good | 250 | 3.21 (0.40) | 0.41 | 3.19 (0.48) | 0.35 | 3.23 (0.52) | 0.49 | 3.11 (0.45) | 0.26 | 3.36 (0.56) | 0.20 |
| Excellent | 221 | 3.22 (0.42) | 0.41 | 3.19 (0.53) | 0.31 | 3.25 (0.53) | 0.51 | 3.14 (0.46) | 0.32 | 3.36 (0.61) | 0.20 |
| *P* value | | 0.081 | | 0.097 | | 0.028 | | 0.349 | | 0.719 | |
| **Academic performance (ELA)** | | | | | | | | | | | |
| A | 283 | 3.29 (0.37) | Ref | 3.26 (0.45) | Ref | 3.35 (0.46) | Ref | 3.15 (0.43) | Ref | 3.45 (0.52) | Ref |
| B | 232 | 3.10 (0.44) | −0.46 | 3.10 (0.54) | −0.33 | 3.06 (0.58) | −0.57 | 3.05 (0.47) | −0.22 | 3.24 (0.63) | −0.37 |
| C/D | 50 | 3.09 (0.53) | −0.49 | 3.02 (0.58) | −0.51 | 3.05 (0.78) | −0.58 | 3.04 (0.54) | −0.23 | 3.37 (0.66) | −0.15 |
| *P* value | | <0.001 | | <0.001 | | <0.001 | | 0.041 | | <0.001 | |
| **Academic performance (Math)** | | | | | | | | | | | |
| A | 268 | 3.28 (0.39) | Ref | 3.26 (0.48) | Ref | 3.32 (0.47) | Ref | 3.15 (0.46) | Ref | 3.43 (0.51) | Ref |
| B | 230 | 3.13 (0.43) | −0.35 | 3.10 (0.53) | −0.31 | 3.13 (0.60) | −0.36 | 3.05 (0.45) | −0.23 | 3.30 (0.63) | −0.22 |
| C/D | 60 | 3.11 (0.50) | −0.40 | 3.11 (0.52) | −0.31 | 3.05 (0.72) | −0.51 | 3.09 (0.50) | −0.14 | 3.24 (0.70) | −0.35 |
| *P* value | | <0.001 | | 0.002 | | <0.001 | | 0.047 | | 0.015 | |

**Notes.**

Abbreviations: *ELA*, English Language Arts; *Math*, Mathematics; *SD*, Standard deviation; *ES*, Effect size; *Ref*, Reference category.

**Table 5** Correlations (Pearson's *r*) among different domains of the Quality of Life in School (QoLS) scale.

|  | Total QoLS | PS | AT | SE | TS |
|---|---|---|---|---|---|
| Total QoLS | – | 0.83 | 0.86 | 0.76 | 0.78 |
| PS |  | – | 0.59 | 0.44 | 0.44 |
| AT |  |  | – | 0.57 | 0.63 |
| SE |  |  |  | – | 0.61 |

**Notes.**

Abbreviations: *PS*, Psychosocial; *AT*, Attitude towards school; *SE*, School environment; *TS*, Teacher–student relationship.

analysis. The remaining three categories (*Teacher–student relationship and school activities*, *Physical environment*, *Positive feelings toward school*) were similar to the ones found in our study but the number of items in each of the four factors differed from the original publication: while Weintraub and Erez reported 8, 5, 11, and 12 items, respectively, in the *Negative feelings towards school*, *Positive feelings towards school*, *Physical environment*, and *Teacher–student relationship and social activities*, we found 12, 8, 8 and 6 items in the corresponding subscales (two items were dropped because of loadings <0.30). Since the authors did not publish the detailed results from their factor analysis we can only speculate that these differences are related to the different cultural contexts that the instrument was used in the two studies. Despite these differences, the structure of the QoLS as determined in this study is consistent with the theoretical framework underlying the QoLS, and in a broader sense also with that of the *School Well-Being Profile*, which covers similar domains as discussed in the Introduction (*Konu & Lintonen, 2006a*; *Weintraub & Bar-Haim Erez, 2009*; *Konu & Lintonen, 2006b*). Future studies in the Canadian context should use the shortened 34-item version with the revised 4-factor structure (Table 2) developed in the current study. The 4-factor structure explained 39% of the variance in our sample of young Canadian students, which is less than the variance reported in the original publication (51 %) (*Weintraub & Bar-Haim Erez, 2009*). The different amount of variance explained by the model may be related to the fact that the finding is from the analysis of the pre-final version of the QoLS scale, which used 40 items. The percentage variance explained was not reported for the factor structure of the *School Well-Being Profile* (*Konu & Lintonen, 2006a*; *Konu et al., 2002*) or any of the other quality of school life scales.

The hypothesis regarding the subscale correlations also showed acceptable results. All four factors had moderate to strong significant correlations with each other and with the total score lending support to its good convergent validity. The results are also in accordance with the previous research and support the fact that each of the QoLS factors contributes to the measurement of the quality of life construct at school. Of note, Weintraub and Erez found a (weak) negative correlation of the *Negative feelings towards school* factor with both the *Physical environment* and *Teacher–student relationship* factor, which indicated that the students with worse feelings about themselves and their school perceived better physical environment and teacher student relationship (*Weintraub & Bar-Haim Erez, 2009*). These observations were hard to justify. By contrast, our study demonstrated no negative correlations amongst different factors imparting support to the convergent validity of the instrument.

The psychometric analysis of the QoLS measure showed desirable results. The scale was found to be acceptable on the tested sample of students since the number of missing item responses was low and the time spent completing the questions was brief. The internal consistency overall and for the QoLS subscales was excellent. The results of known-group comparison demonstrated that the QoLS was sufficiently sensitive to discriminate between different subgroups by gender, grade and academic performance, in accordance with our a priori hypotheses for the measure. Girls had higher scores than boys with the exception of the *Psychosocial* subscale. (*Konu & Lintonen, 2006*; *Konu & Lintonen, 2006b*) also found higher scores among girls across all domains of the *School Well-Being Profile* except *Health status*, which is similar in scope to the "Psychosocial" subscale in the QoLS. The effect sizes for the gender differences in QoLS scores were fairly small, which is in keeping with existing literature (*Mok & Flynn, 2002*; *Malin & Linnakyla, 2001*; *Konu & Lintonen, 2006*; *Løhre, Lydersen, Vatten, 2010*; *Kong, 2008*). The decrease of the QoLS score with age is also well described in the literature (*Kong, 2008*; *Weintraub & Bar-Haim Erez, 2009*; *Samdal et al., 1998*; *Konu & Lintonen, 2006*; *Park, 2005*). The largest effects were seen in the *Teacher–student relationship* and *School environment* subscale but surprisingly there was no change in the *Attitude towards school* subscale. We also found very strong positive associations of academic performance with QoLS scores with effect sizes of up to 0.6 standard deviation units. This association has been described before by others (*Mok & Flynn, 2002*; *Murray-Harvey, 2010*) but due to the cross-sectional design of those studies and the present one, no statement is possible about directionality of the association. Overall and subscale scores for the QoLS increased with parent-reported health status but the differences were only significant for the *Attitude towards school* subscale but not for the *Psychosocial* subscale, which contains a number of health-related items. Possibly, the parents' assessment of their child's health status was more reflective of physical and mental health problems that require medical attention, while the *Psychosocial* subscale captured subclinical problems that the parents may not be aware of.

The concept of quality of school life also closely relates to the health promoting schools (HPS) approach (*Deschesnes, Martin & Hill, 2003*; *World Health Organization, 1997*). HPS is an internationally recognized framework for supporting improvements in students' educational outcomes while addressing school health in a planned, integrated and holistic way. Similarly, the quality of school life approach also intends to assess the students' health, educational and social outcomes in an integrated fashion that encompasses different aspects of school environment. Hence, the QoLS instrument or other measures of quality of school life may be used to evaluate outcomes of HPS interventions, which have documented challenges with outcome assessment due to its varied implementation in different schools or settings (*Keshavarz et al., 2010*; *Mūkoma & Flisher, 2004*). In fact, some items on the *School environment* and *Teacher–student relationship* domains of the QoLS instrument align with the environmental and curriculum domain of HPS but further research is needed to examine use of the QoLS tool in such a setting. The sensitivity of the instrument in picking up differences in gender, age, and academic performance needs to be taken into account when QoLS scores are compared e.g., between schools with and without

an HPS program as such differences may lead to a spurious estimates of the effect of an intervention.

The limitations of the study included an inability to perform test–retest reliability and test responsiveness of the scale due to the cross sectional nature of the study design. However, the floor and ceiling effects were computed and found to be acceptable except for the teacher–students relationship domain, which had significant ceiling effect. These findings may have considerable implications for the planning and evaluation of effectiveness of the school interventions. Another limitation may be the study setting in a rural, socioeconomically disadvantaged region of Canada, which may not be representative of more urban areas. Strengths of our study include a population-based study design and the availability of academic performance and socio-demographic information on the participants, which allowed for known groups comparisons.

## CONCLUSIONS

Our study has demonstrated that the QoLS scale is a reliable and valid tool for quality of school life assessment in Canadian elementary school children. Quality of school life is positively associated with academic performance in elementary school students. Future studies in the Canadian context should use the shortened 34-item version with the revised 4-factor structure developed in the current study. The QoLS scale may be used as a student-level outcome in the evaluation of HPS interventions but more research is needed to evaluate the instrument in that context.

### Funding

This work was supported by the Canadian Institutes of Health Research (CIHR) (FRN 127082). Sara Kirk holds a CIHR Canada Research Chair in Health Services Research. Jessie-Lee McIsaac was supported by a postdoctoral fellowship from KT Canada and CIHR as well as a Bright Red Graduate Research Award from the Heart and Stroke Foundation of Canada. The funders had no role in study design, data collection and analysis, decision to publish, or preparation of the manuscript.

### Grant Disclosures

The following grant information was disclosed by the authors:
Canadian Institutes of Health Research: FRN 127082.
KT Canada.
Heart and Stroke Foundation.

### Competing Interests

Stefan Kuhle is an Academic Editor for PeerJ. The other authors declare there are no competing interests.

### Author Contributions

- Satvinder Ghotra analyzed the data, wrote the paper, prepared figures and/or tables, reviewed drafts of the paper.

- Jessie-Lee D. McIsaac conceived and designed the experiments, wrote the paper, reviewed drafts of the paper.
- Sara F.L. Kirk conceived and designed the experiments, reviewed drafts of the paper.
- Stefan Kuhle conceived and designed the experiments, analyzed the data, wrote the paper, prepared figures and/or tables, reviewed drafts of the paper.

### Human Ethics

The following information was supplied relating to ethical approvals (i.e., approving body and any reference numbers):

Ethics approval for this study was obtained from the Health Research Ethics Boards at the Dalhousie University (Protocol # 2013-3094). The permission for data collection was also granted from the participating school board and individual schools. Written informed consent was obtained from the parents of all children included in the study.

### Data Availability

Raw data are not provided for publication as the ethics agreement for the study does not allow distribution of the data to anyone but the study investigators.

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
