# Peer review of "Validation of the “Quality of Life in School” instrument in Canadian elementary school students"

_PeerJ, doi:10.7717/peerj.1567_

## Round 0.1 · original submission · Major Revisions

Please ensure you address all of the reviewers' comments in the revised article, and include a point by point description of the changes that have been made.

Reviewer 1 ·

Basic reporting

Overall, the Introduction is well written. The study's objective is clearly stated, and the authors explain the importance of measuring school QOL, in general, and specifically in health promoting schools. However, there are certain areas that require more elaboration.
1. The authors write in the Introduction that "There are a few QSL measures available." and state in a sentence why they did not select these questionnaires. However, only in the Discussion do they actually review the measures and provide a more in-depth explanation for their rationale. In my opinion, the review presented in the Discussion (from the 3rd paragraph on p.7: "There are a number of tools available…" until row 195 on p.8: "…alongside with other instruments in a large survey of young children.") should be moved to the Introduction.

2. Part of the validation process of the QoLS is by comparing genders and age groups. This process has a good rationale in the literature, but the authors do not provide this information. It is suggested that the authors present a short review of what is known in the literature about gender and age effect on students' perception on their QOL.

3. In the Results section (row 137 of p.6) the authors state: "Completion of the scale questions took about ten minutes." This sentence should be moved to the Methods section, because it describes the procedure rather than the results.

4. In the Discussion (row 237, p.10) the authors wrote "Of note, Weintraub et al. found…" It should be Weintraub and Erez found…"

Experimental design

The purpose of this study is clearly stated. The design is appropriate to meeting the purpose. The Material and Methods section includes the necessary information in a concise and clear manner.

Validity of the findings

1. Participants' description is inclusive except for the participants' age. Although grade-levels are stated, due to the differences among countries in terms of the age that students go to school, it would be helpful if the authors provide this information.

2. Although demographic data is presented in the tables, it is helpful to the readers (and customary) to summarize the major findings and trends in the text (unless this specific journal does not adhere to this custom).

3. Data analysis is appropriate and congruent with the study's objective. I do, however, believe that in comparing differences between genders or grade-levels with respect to the different categories of the QOLS, the authors should have employed MANOVA rather than t-tests or ANOVA, due to the tact that these categories are related (as the authors themselves reported).

4. In describing the correlation among the categories of the QoLS (Table 4) the authors should add the 'p' values of the correlations.

5. The conclusions are stated and supported by findings of previous studies, except with respect to the findings relating to gender and grade-level effect of QoLS. The authors did mention one study (row 224, p.9), however it is recommended to provide a review of a wider range of studies so as to support this study's findings.

Reviewer 2 ·

Basic reporting

Abstract:
In the abstract, the authors introduce the Health Promoting Schools, which aim to improve students' educational and health outcomes in school in an integrated and holistic way. Although I see the relevance of linking the QoLS instrument to this, HPS are not the core focus of the study. It seems that the study presented here is part of a larger project which includes studying the effects of HPS on students' perception of their quality of school life. For the validation of the instrument, this broader context receives, in my view, too much attention, particularly because the results (e.g. regarding the different domains) is not interpreted in light of the characteristics of HPS.
Another point of discussion is that a conclusion is drawn about the construct validity of the instrument, but I would suggest the readers to nuance this statement. Construct validity based on quantitative data only is a limited view on construct validity. Triangulation, that is, cross-validating the findings with other types of instruments (e.g. qualitative interview data on a smaller sample) would result in additional support for the construct validity. So yes, the authors did find some information about construct validity, but it is, in my view, not 'evidence' of construct validity.

Introduction:
In the introduction, there are various definitions of QSL used. The authors refer to well-being, the degree of satisfaction or dissatisfaction, students' attitude and feelings towards school. My main problem with the paper in its current state is that the construct (and the four domains within the QSL) are not discussed from a theoretical point of view. Which theoretical grounding lies behind the construct? Why and how did the developers of the instrument come to their four domains? How does the construct relate to similar constructs from different conceptual frameworks? Once this is explained in more detail, the authors would be able to ground their hypotheses on the differences between gender groups, age groups, et cetera on theory. At the moment, there is no argumentation included for the (direction of) the hypotheses, or why comparing these groups is important (rather than comparing them on other characteristics, e.d. social-emotional factors).
Another topic is that the validity can only be established in a specific context (in this case the Canadian context), and it would be helpful for the international audience to include some information on this (school) context in Canada. Also facts about the overall quality of school, the quality of life, school characteristics and student population characteristics in Canada would help readers to understand the issue of quality of school life to a larger extent.

Experimental design

I would place information on the response rates in the method section rather than the results section. Please also include information on the representativeness of the sample here, and whether the missing data was missing at random (or selective).
The QoLS questionnaire has some typical characteristics. I wonder why 'negative feelings toward school' and 'positive feelings toward school' are not in fact endpoints of one single continuum. Again, theoretically, this might be a misinterpretation, but this is not explained in the manuscript.
Adaptations of the questionnaire (e.g. adding an additional item and adapting it to the Canadian context) needs some additional clarification as well. How was this done and why?
I wonder (just a suggestion) whether a higher-order model would make sense here, the authors could take a look at this if they agree that quality of school life is the higher order factor with 4 lower order factors.
The response categories (a 4-point Likert scale) are in fact ordinal, not continuous. Although I understand that it is common to treat Likert scales as continuous scales, it would be good to discuss this decision in the paper (maybe in the limitations section), or look into IRT analyses to analyze ordinal scales.
Hot deck imputation is not frequently used as far as I know; please add some information on this method.

Validity of the findings

Results:
Clear descriptions, no comments on this part except for the tables. Please include factor loadings (as numbers) in a table, that is more commonly used and much easier to interpret. For the readers' convenience, order the items according to the 4 domains. This makes is easier to directly see from the table that the results were and which items relate to which domain.
What did the authors do with cross-loadings in the model?
Please add effect sizes for, for example, gender differences, because you have quite a large sample (so almost everything will be statistically significant)

Discussion:
The whole section on other instruments should, in my view, be placed in a separate section, maybe in the theoretical framework. The discussion should be about the authors' own results, and about reflecting on these findings in light of the theory discussed earlier.
The authors state that the QSL structure is consistent with the theoretical definition and literature on QSL, but this is not explained clearly.
In the discussion, HSP is a topic again, but this is detached from the rest of the discussion.
To some extent the authors found differences in comparison with the developers of the instrument. Again, try to come up with theoretical explanations for these differences (e.g. different societal context). This also applies to the explained variance, why did the authors found a lower percentage of explained variance among their sample?
Finally, with regard to the limitations: one could split the sample (randomly), conduct exploratory factor analysis on half of the data, and then perform confirmatory factor analysis on the second half of the data. That way you have a sort of test-retest reliability, and also a stronger case for the validity questions.
The instrument probably could benefit from some improvements with regard to the scales and/or the items. This should be addressed in the discussion.
Adding limitations with regard to representativeness of the sample (and also addressing this when interpreting the findings) would strengthen the paper as well.

---

## Round 0.2 · Minor Revisions

Thank you for your hard work in revising this paper, which is much improved. Please see the one very minor comment from the reviewers, after which we will be happy to accept your paper.

Reviewer 2 ·

Basic reporting

The authors did an excellent job in addressing all comments from the previous review round. I found only one minor error: in Table 1 the percentages are missing for the different age groups.

Experimental design

No new comments.

Validity of the findings

No new comments.

Additional comments

In my view, the paper is now ready for publication (see note about small error in Table 1).

---

## Round 0.3 · accepted · Accept

Thank you very much for your speedy revisions

---

## Author Rebuttal · Round 0.3

Dear Laura,

Please find attached the revised version of our manuscript *"Validation of the "Quality of Life in School" instrument in Canadian elementary school students"*.

The missing percentages have been added to Table 1. We also fixed some minor wording issues and corrected the plus/minus signs for the effect sizes in the "Grade" row in Table 4.

Best wishes,
Stefan

Stefan Kuhle, MD, MPH, PhD
Epidemiologist, Perinatal Epidemiology Research Unit
Assistant Professor, Depts of Pediatrics, Obstetrics & Gynaecology, and Community Health & Epidemiology, Dalhousie University
Email: stefan.kuhle@dal.ca
Website: http://medicine.dal.ca/peru